# Effects of Lockdown for COVID-19 Pandemic on Chronic Kidney Disease Progression in Children with Congenital Anomalies of the Kidney and Urinary Tract: A Retrospective Pilot Study

**DOI:** 10.3390/children10010123

**Published:** 2023-01-06

**Authors:** Pier Luigi Palma, Anna Di Sessa, Antonio Paride Passaro, Eleonora Palladino, Giuseppe Furcolo, Annalisa Barlabà, Giulio Rivetti, Maeva De Lucia, Emanuele Miraglia del Giudice, Stefano Guarino, Pierluigi Marzuillo

**Affiliations:** 1Department of Woman, Child and of General and Specialized Surgery, Università degli Studi della Campania “Luigi Vanvitelli”, 80138 Naples, Italy; 2Unità Operativa Complessa di Pediatria e Pronto Soccorso Pediatrico, AORN Moscati, 83100 Avellino, Italy

**Keywords:** congenital anomalies of the kidney and urinary tract, chronic kidney disease, COVID-19, obesity, children

## Abstract

The coronavirus disease 2019 (COVID-19) pandemic changed adults and children’s lifestyle. We focused our attention on children affected by chronic kidney disease (CKD) due to congenital abnormalities of kidney and urinary tract (CAKUT) and their behavior during the lockdown. Our aims were to evaluate the incidence of CKD progression within 6 months after the end of the first Italian lockdown and the factors associated to it. CKD progression was defined by the transition to higher CKD stage or by the drop in estimated glomerular filtration rate by a 25% or more for patients belonging to CKD stages 1 and 2. We retrospectively selected 21 children with CAKUT and CKD ≥ stage 1 observed within 3 months before and 6 months after the first Italian lockdown. We called them by phone and asked them about their lifestyle before and during lockdown focusing on physical activity, screen time, sweet/candies/sugar-sweetened beverages eaten/drunk and adherence to the Mediterranean diet (MD) (through KIDMED questionnaire). We calculated and analyzed the delta between the pre- and post- lockdown observation of all collected parameters (clinical and biochemical parameters and questionnaires scores). Analyzing the overall cohort, we found significantly increased mean BMI and mean screen time and significantly lower mean physical activity time in post- compared with pre-lockdown observations. Eleven out of twenty-one patients (52.4%) had a worsening of CKD. These patients presented higher delta of levels of uric acid and microalbuminuria and showed minor adherence to the MD and declared to have consumed more sweets or candies or sugar-sweetened beverages/week during the lockdown with a tendentially major increment of BMI compared with patients not presenting CKD progression. In conclusion, the lockdown for COVID-19 pandemic determined increase of BMI in all enrolled patients due to a “forced” negative lifestyle. About half of these patients presented CKD progression. This progression was associated to less adherence to the MD and major consumption of sweets or candies or sugar-sweetened beverages.

## 1. Introduction

The coronavirus disease 2019 (COVID-19) pandemic has had big repercussions on people’s life. Even children have suffered for this pandemic at several levels. Although the infection itself was reported not to be dangerous in children, with a very low incidence of moderate to severe symptomatic disease [1], it has been reported, afterwards, to cause the Multisystem Inflammatory syndrome in Children (MIS-C), with a needing of intensive care admission up to 68% of cases [2]. Long COVID has also been described in children with mild to severe symptoms [3]. Moreover, various mental health problems among children have been identified such as an increase of anxiety, depression, loneliness, stress, fear, tension, anger, fatigue, confusion, and worry [4].

In Italy the first lockdown lasted 2 months, from March to April 2020, with looser measures at the beginning of May 2020. Evidence indicates that during lockdown there has been an increase of Body Mass Index (BMI) in adults [5] and children as well [6]. Increased stress, irregular mealtimes, less physical activity and increased screen time are all contributing factors to the BMI increase. In addition, children and adolescents spent more time away from structured school settings which is also associated with weight gain [7].

Overweight and obesity can negatively affect renal function already in childhood [8,9,10]. Moreover, evidence indicate that obesity plays a role in the progression of the chronic kidney disease (CKD) [11].

We focused our attention on children affected by CKD and their behavior during the first Italian lockdown, supposing that lifestyle changes during lockdown could have negatively influenced the estimated glomerular filtration rate (eGFR). As the progression of CKD due to glomerulonephritis is different from the one due to congenital abnormalities of kidney and urinary tract (CAKUT), we limited the study to the latter subjects. Our aims were to evaluate the incidence of CKD progression within 6 months after the end of the first Italian lockdown and the factors that could have contributed to it.

## 2. Materials and Methods

To perform this study, we retrospectively selected all the patients with CKD (≥stage 1 CKD [12]) who regularly undergo to follow-up in our clinic. We retrospectively identified (between December 2021 and May 2022) the children (*n* = 34) observed within 3 months before the start of the first Italian lockdown (pre-lockdown observation between December 2019 and February 2020) and observed again for a regular follow-up within 6 months after the end of that lockdown (post-lockdown observation between May 2020 and November 2020). After the retrospective identification of the eligible patients, their parents were contacted by phone (between January and June 2022) and were submitted to questionnaires about usual lifestyle of their children before and during the first Italian lockdown.

Patients denying their consent to be included into the study (*n* = 2) or denying their consent to answer to the telephonic interview (*n* = 3) were excluded from the study. Patients contracting COVID-19 infection (*n* = 5) and urinary tract infection (UTI) (*n* = 3) between the two observations were excluded. Therefore, we enrolled 21 patients.

Our Ethical committee approved the study (0010396/i). Mainly we asked about how many hours for week their children spent in front of a screen (tablet, computer, videogames, television and so on), how many sweets or candies or sugar-sweetened beverages they ate/drank in a week, how many hours for week of physical activity they practiced, for both pre-lockdown and lockdown period (please see below for more details).

We tried to prevent recall bias by arranging with each parent the date and hour to make the interview and limiting the duration of the phone interview to a maximal duration of 30 min. The interview was in Italian language.

### 2.1. Time Spent in Front of a Screen

The asked questions to parents were based on the Health Behavior in School-aged Children Survey [13] and on previously validated questions [14]. This item included every-day time spent in front of a computer, television, mobile phones or video-games by children both in pre-lockdown period and during lockdown. We used the following questions: “How many hours per day...” does your child usually watch television (including videos on a smartphone) during her/him free time?... Does your child usually play video games on a computer or a console during her/him free time?... Does your child usually use a computer to participate in forum discussions (“chats”) or surf the Internet or send e-mails, or to do homework during her/him free time?... How much time does your child spend with your friend(s) talking on the phone, or sending text messages?

The total amount of hours spent by children in front a screen was calculated both in pre-lockdown and during lockdown period. However, the hours spent in front a computer for the distance learning, usually adopted in Italy during lockdown, were not accounted.

### 2.2. Physical Activity

The physical activity questionnaire was modified accordingly to the children’s physical activity questionnaire [15]. We asked how many times/week and how many hours/minutes lasted each sport activity both in and outside school activities and we calculated the total hours of physical activity in both pre-lockdown and lockdown period.

### 2.3. Sweets, Candies and Sugar-Sweetened Beverages

Parents were asked to report on the frequency with which their children consumed sweet and beverages both before and after lockdown period by choosing one of the following options for each item: “consumed sweets, candies and sugar-sweetened beverages less than once a month”, “consumed less than once a week”, “consumed once a week”, “consumed 2 to 3 times a week”, “consumed 4 to 6 times a week”, “consumed once a day”, “consumed twice a day” and “consumed more than three times a day” [16].

### 2.4. Adherence to the Mediterranean Diet

We evaluated the adherence to the Mediterranean diet through the Mediterranean Diet Quality Index for Children and Adolescents (KIDMED) questionnaire [17,18]. The quality of Mediterranean diet (MD) was defined based on the KIDMED score: ≤3, very-low-quality diet; 4–7, need to improve the food pattern to adjust it to the Mediterranean one; ≥8, optimal Mediterranean diet.

### 2.5. Control Group

We also retrospectively selected an historical cohort of 21 patients matched for age and sex evaluated between 2004 and 2007 who underwent two follow-up visits with an interval time between the two visits of 11–14 months. Because this cohort was historical, we did not submit these patients to questionnaires.

### 2.6. Retrospectively Collected Clinical and Biochemical Data

All the data were retrospectively collected from the clinical charts between December 2021 and May 2022. At every follow-up visit of patients with CKD we routinely measure and then we collected the following data: height, weight, BMI with its Standard Deviation Score (SDS), blood pressure, serum creatinine (mg/dL), sodium (Na), potassium (K), phosphorus, bicarbonates (HCO3^−^), ferritin, total cholesterol, triglycerides, parathormone, vitamin D, glycemia, glutamate-pyruvate transaminase (GPT), glutamate-oxalacetate transaminase (GOT), a complete blood count with neutrophils, white blood cells (WBC), platelets, red blood cells (RBC), hemoglobin, urinalysis, microalbuminuria and urinary protein/creatinine ratio (UPr/Cr). Blood pressure was measured by automated oscillometric device with appropriate cuff size, taking in consideration the lowest value among three measurements [13]. Serum creatinine was measured by Jaffe method (methodology: Alkaline Picrate, Abbott catalog no. 7D64-20) by using the Architect c16000 automated analyser (Abbott Diagnostics Inc., Park City, IL, USA), as usually made in our center [19], with eGFR calculated by using the original Schwartz equation [20]. We routinely perform urine culture in CAKUT patients with urinary symptoms or fever with no evident cause to rule out UTI. Regarding the historical cohort, we decided to collect from the clinical charts only age, height, sex and serum creatinine levels of patients to evaluate the eGFR and the CKD progression only. In fact, during the lest 15–16 years the techniques of biochemical dosages and the life-style changed, so further and more detailed analyses could have been invalidated by several factors.

### 2.7. Patients’ Classification

We divided our population into two groups: children who had progression of CKD, and children who had not. CKD progression was defined by the transition to higher CKD stage [21]. Moreover, because we also evaluated children with stage 1 CKD that is defined by eGFR > 90 mL/min/1.73 m^2^ and stage 2 CKD that is defined by eGFR between 89 and 60 mL/min/1.73 m^2^ and then with a wide range of variability of eGFR allowing to remain in the same CKD stage also with a significant reduction of eGFR (eGFR variation > 25% [21]), we also considered as CKD progression the drop in estimated glomerular filtration rate by a 25% or more for patients belonging to CKD stage 1 and 2. This drop in of 25% or more, in fact, is considered significant in the KDIGO guidelines [21]. Finally, to define the progression, we also retrospectively evaluated from the clinical charts the persistence of the worsening in eGFR levels at least for 3 months.

### 2.8. Post-Hoc Power Calculation

On the basis of a rate of CKD progression in our historical cohort of 19%, considering the rate of CKD progression of 52.4% in the present cohort, the calculated Post-Hoc power, with an alpha of 0.05, was 93.6%.

### 2.9. Statistical Analysis

For all the collected parameters (both clinical and biochemical and scores to questionnaires) we calculated and analyzed the delta between the post- and pre-lockdown values, and we evaluated differences in these values comparing patients with and without CKD progression.

*p* values < 0.05 were considered statistically significant. Differences for continuous variables were analyzed with the independent-sample *t*-test for normally distributed variables and with the Mann–Whitney test in case of non-normality. Qualitative variables were compared by using Fisher exact test. Moreover, we evaluated the collected parameters of the population comparing the pre- and post-lockdown observation. For these latter analyses, we used the paired *t*-test for parametric data and the Wilcoxon matched paired test for non-parametric data.

The data are presented as mean ± SDS for normally distributed variables and as median and interquartile range (IQR) for non-normally distributed variables. The Stat-Graph XVII software for Windows was used for all statistical analyses.

## 3. Results

Twenty-one patients with CAKUT-related CKD were enrolled. At the pre-lockdown observation seven out of twenty-one patients (33.3%) presented stage 1 CKD, twelve (57.1%) stage 2 CKD, one (4.8%) stage 3 CKD and one (4.8%) stage 4 CKD. No patient presented stage 5 CKD.

The mean age of enrolled patients was 10.63 ± 4.8 years. Five patients presented an age between 3 and 5 years, 5 patients between 6 and 10 years, 6 between 10 and 15 years, and 5 between 16 and 18 years. Sixteen out of twenty-one (76.2%) were of male sex. No patients suffered UTI during the study period. Eleven patients had a worsening of CKD (52.4%), ten of which (90.9%) were of male sex.

Among the 11 patients experiencing CKD worsening, two passed from stage 1 to stage 2, one passed from stage 2 to stage 3 while eight patients presented at least 25% reduction in eGFR levels but remained in the same CKD stage.

In all patients, we found higher BMI and BMI-SDS levels, lower physical activity (hours/week) and KIDMED score and higher screen time (hours/week) and sweets/candies/sugar-sweetened beverages consumption at post- compared with pre-lockdown observation (Table 1).

The age and prevalence of pubertal children both at pre-lockdown and post-lockdown follow-up visit were similar for patients with and without CKD progression (Table 2).

As expected, patients with CKD progression presented higher delta of creatinine and eGFR levels compared with patients without CKD progression (Table 2).

Moreover, patients with CKD progression presented higher delta of KIDMED score and of uric acid and microalbuminuria levels and showed a tendentially major increment of BMI compared with patients without CKD progression (Table 2).

In addition, patients with CKD progression declared to have consumed more sweets or candys/week during the lockdown (3.5 ± 3.4 vs. 0.7 ± 0.9; *p* = 0.02) (Table 2).

### Comparison with the Historical Cohort

In the historical cohort, the mean age (10.8 ± 5.1 SDS), the percentage of patients of male sex (15/21; 71.4%) and of pubertal patients (10/21; 47.6) were similar to those of the cohort enrolled for this study (*p* = 0.95, 0.72 and 0.76, respectively). In the historical cohort 4 out of 21 (19%) of patients showed CKD progression compared with the progression CKD rate of 52.4% of the present cohort (*p* = 0.05).

## 4. Discussion

This is the first study investigating the effect of lockdown for COVID-19 on the progression of CKD in children affected by CAKUT-related CKD. We found a common worsening of lifestyle during lockdown for COVID-19 pandemic in children affected by CAKUT-related CKD, sustained by less adherence to the MD, less physical activity, and more screen time and sweets/candies/sugar-sweetened beverages consumption. This is in line with Dor-Haim et al., who found that 70% of Israelis reduced their physical activity and 55% of them had a gain weight [22,23]. In our population, these changes were associated to a CKD worsening, indeed eleven out of twenty-one patients with CKD had a worsening of their condition. In particular, the minor adherence to the MD and the major consumption of sweets/candies/sugar-sweetened beverages were more prevalent among patients presenting compared with those not presenting CKD progression.

The MD has several benefits. Menotti et al. reported lower coronary heart disease mortality in Mediterranean countries compared to northern European countries or the USA [24]. On this line, Chauveau et al. summarized the potential positive effects (controlled protein intake, lower amount of dietary sodium, phosphate, potassium and lower intake of saturated fatty acids versus rich intake of monounsaturated fatty acids) of MD in patients affected by CKD [25].

We found an increase of BMI in all patients, which is commonly described during COVID-19 pandemic [5,6]. The BMI increase could have deleterious impact on kidney function [26] while weight-loss may improve kidney function in children with non-glomerular CKD [11]. The worsening in eGFR in case of overweight or obesity may be explained by worse metabolic parameters such as lipids and insulin sensitivity, which may impact kidney function [8].

Interestingly, we observed a higher delta of uric acid in patients with CKD progression. This could be explained in part by the fact that uric acid is mostly cleared by kidneys, with an increase of serum levels secondary to GFR worsening. On the other side uric acid could be itself a contributing factor to the CKD progression by causing hypertension and/or cardiovascular disease [27].

Limitation of our study is represented by the retrospective design and by the low number of enrolled subjects. Despite this, the power was acceptable and we were able to detect significant differences comparing patients with and without CKD progression. Anyway, we can not exclude the presence of other non-identified differences because the limited sample-size of this study. Another limitation could be represented by the use of questionnaires by phone and by the recall bias. However, we tried to limit it by arranging with each parent the appointment for the interview and limiting the duration of this interview to max 30 min.

Finally, the generalizability of our study is limited due to the retrospective nature and the limited sample size. This study, however, should be mainly considered as only another proof of concept of the impact that the lifestyle can have on the kidneys’ health.

We think and hope that the experience of a new pandemic with a new lockdown will never present again. For this reason, a similar larger study it will be difficult to obtain. The recent lockdown for COVID-19 pandemic has provided evidence about the effects on the CKD progression of an abrupt change toward a negative lifestyle as shown in the present study. In case of future pandemic, however, it should be carefully evaluated if the lockdown for patients with CKD could be really beneficial. This because, on one hand, the lockdown certainly reduced the risk of infection but, on the other hand, it could predispose to negative lifestyle with negative impact on CKD progression. As future perspective, it could be interesting to evaluate in a large international cohort of patients with CKD due to CAKUT the effects of the life style and of eating habits on the CKD progression in childhood in order to improve the conservative management of CKD starting since childhood.

In addition to the direct effects (due to the COVID-19 infection) of the COVID-19 pandemic on people’s health, also a lot of indirect effects (due to the “forced” changes in the life-style, social distancing, etc.) have been observed [1,2,3,4]. In this study, we analyzed some indirect COVID-19 effects in patients with CKD due to CAKUT. All enrolled patients presented increased BMI, screen time and sweets/candies/sugar-sweetened beverages consumption and lower physical activity time during the first Italian lockdown for COVID-19 pandemic. About half of these patients presented CKD progression. This progression was associated to less adherence to the MD and major consumption of sweets or candies or sugar-sweetened beverages.

## Figures and Tables

**Table 1 children-10-00123-t001:** Clinical and biochemical characteristics of all patients at pre- and post-lockdown observation. For continuous normally distributed variables mean ± SDS is shown, while for non-normally distributed variables median and interquartile range are shown.

	Pre-Lockdown Observation	Post-Lockdown Observation	p (Used Test)
Pubertal, No. (%)	9 (42.9)	12 (57.1)	0.54 (chi-square test)
Stage 1 CKD, No. (%)	7 (33.3)	5 (23.8)	0.73 (chi-square test)
Stage 2 CKD, No. (%)	12 (57.1)	13 (61.9)	0.99 (chi-square test)
Stage 3 CKD, No. (%)	1 (4.8)	2 (9.5)	0.99 (Fisher’s exact test)
Stage 4 CKD, No. (%)	1 (4.8)	1 (4.8)	0.99 (Fisher’s exact test)
Stage 5 CKD, No. (%)	0 (0)	0 (0)	0.99 (Fisher’s exact test)
BMI, Kg/m^2^, mean (SDS)	21.03 (4.71)	22.81 (5.21)	<0.01 (Paired *t*-test)
BMI-SDS, median (IQR)	0.85 (1.29)	1.53 (1.13)	<0.01 (Wilcoxon matched paired test)
SBP, mmHg, mean (SDS)	106.43 (12.92)	106.95 (11.36)	0.98 (Paired *t*-test)
DBP, mmHg, mean (SDS)	65.81 (7.63)	66.10 (6.32)	0.24 (Paired *t*-test)
Creatinine, mg/dL, median (IQR)	1.01 (0.55)	1.04 (0.73)	0.06 (Wilcoxon matched paired test)
eGFR, ml/min/1.73 m^2^, mean (SDS)	82.39 (21.99)	77.91 (22.04)	0.05 (Paired *t*-test)
UPr/UCr, median (IQR)	0.2 (0.19)	0.21 (0.28)	0.33 (Wilcoxon matched paired test)
Uric acid, mg/dL, mean (SDS)	5.78 (1.79)	6.01 (1.74)	0.4 (Paired *t*-test)
Phosphorus, mg/dL, mean (SDS)	4.48 (0.94)	4.38 (0.84)	0.51 (Paired *t*-test)
Parathormone, pg/mL, median (IQR)	22.8 (15.5)	24.4 (19.9)	0.41 (Wilcoxon matched paired test)
Vitamin D, ng/mL, mean (SDS)	26.86 (12.16)	26.48 (10.76)	0.78 (Paired *t*-test)
Neutrophils, ×10^3^/µL, mean (SDS)	4.18 (1.29)	4.15 (1.29)	0.93 (Paired *t*-test)
White Blood Cells, ×10^3^/µL, median (IQR)	7.41 (2.62)	7.58 (1.65)	0.12 (Wilcoxon matched paired test)
Platelets, ×10^3^/µL, mean (SDS)	294 (76.50)	268 (76.00)	0.17 (Paired *t*-test)
Microalbuminuria, mg/L, median (IQR)	16.0 (43)	45.0 (91.0)	<0.01 (Wilcoxon matched paired test)
Glycemia, mg/dL, mean (SDS)	78.44 (11.95)	80.89 (8.43)	0.42 (Paired *t*-test)
GOT, U/L, mean (SDS)	26.29 (7.78)	23.2 (6.79)	0.51 (Paired *t*-test)
GPT, U/L, median (IQR)	18.0 (0.8)	18.5 (0.8)	0.06 (Wilcoxon matched paired test)
Na^+^, mEq/L, mean (SDS)	138.62 (1.80)	139 (1.84)	0.45 (Paired *t*-test)
K^+^, mEq/L, mean (SDS)	4.61 (0.34)	4.43 (0.40)	0.06 (Paired *t*-test)
Cl^−^, mEq/L, mean (SDS)	105.95 (2.60)	106.10 (2.07)	0.79 (Paired *t*-test)
Ferritin, µg/L, median (IQR)	27.5 (22.5)	29.0 (26.0)	0.59 (Wilcoxon matched paired test)
Total cholesterol, mg/dL, mean (SDS)	159. 67 (26.32)	169.25 (27.87)	0.17 (Paired *t*-test)
Triglycerides, mg/dL, median (IQR)	81.0 (38.0)	94.0 (86.0)	0.12 (Wilcoxon matched paired test)
KIDMED score, mean (SDS)	6.57 (2.91)	5.81 (2.48)	0.02 (Paired *t*-test)
Physical activity, hours/week, mean (SDS)	7.90 (4.49)	3.71 (4.08)	<0.01 (Paired *t*-test)
Screen time, hours/week, mean (SDS)	10.10 (5.05)	26.02 (10.54)	<0.01 (Paired *t*-test)
Sweets/Candies, n/week, mean (SDS)	6.90 (3.16)	9.10 (4.94)	<0.01 (Paired *t*-test)

Abbreviations: BMI, body mass index; CKD, chronic kidney disease; DBP, diastolic blood pressure; GOT, glutamate-oxalacetate transaminase; GPT, glutamate-pyruvate transaminase; eGFR, estimated glomerular filtration rate; IQR, interquartile range; KIDMED, Mediterranean Diet Quality Index for Children and Adolescents questionnaire; SBP systolic blood pressure; SDS, standard deviation score; UPr/Cr, urinary protein/creatinine ratio.

**Table 2 children-10-00123-t002:** Clinical and biochemical characteristics of patients presenting or not worsening of chronic kidney disease. For continuous normally distributed variables means ± SDS are shown, while for non-normally distributed variables median and interquartile range are shown.

	Worsening No. = 11	Not Worsening No. = 10	p (Used Text)
Age at observation pre-lockdown, yr	10.78 (4.19)	10.47 (5.9)	0.89 (*t*-test)
Pubertal pre-lockdown, No. (%)	5 (45.4)	4 (40.0)	0.99 (Fisher’s exact test)
Pubertal post-lockdown, No. (%)	7 (63.6)	5 (50.0)	0.7 (chi-square test)
Pre-lockdown stage 1 CKD, No. (%)	4 (36.4)	3 (30.0)	0.99 (Fisher’s exact test)
Pre-lockdown stage 2 CKD, No. (%)	7 (63.6)	5 (50.0)	0.67 (chi-square test)
Pre-lockdown stage 3 CKD, No. (%)	0 (0)	1 (10.0)	0.48 (Fisher’s exact test)
Pre-lockdown stage 4 CKD, No. (%)	0 (0)	1 (10.0)	0.48 (Fisher’s exact test)
Pre-lock-down stage 5 CKD, No. (%)	0 (0)	0 (0)	0.99 (Fisher’s exact test)
Male sex, No. (%)	10 (90.91)	6 (60%)	0.15 (chi-square test)
Δ BMI, Kg/m^2^, mean (SDS)	2.4 (3.6)	1.1 (1.2)	0.28 (*t*-test)
Δ BMI-SDS, mean (SDS)	1.05 (0.71)	0.5 (0.52)	0.06 (*t*-test)
Δ SBP, mmHg, mean (SDS)	0.82 (9.67)	0.2 (9.76)	0.89 (*t*-test)
Δ DBP, mmHg, mean (SDS)	2.55 (6.64)	−2.2 (10.36)	0.22 (*t*-test)
Δ Creatinine, mg/dL, mean (SDS)	0.18 (0.15)	−0.07 (0.15)	<0.01 (*t*-test)
Δ eGFR, ml/min/1.73 m^2^, mean (SDS)	−3.4 (4.6)	11.62 (8.58)	<0.01 (*t*-test)
Δ UPr/UCr, median (IQR)	−0.02 (0.17)	0.05 (0.11)	0.10 (Mann-Whitney-test)
Δ Uric acid, mg/dL, mean (SDS)	0.9 (0.44)	−0.49 (1.5)	<0.01 (*t*-test)
Δ Phosphorus, mg/dL, mean (SDS)	0.04 (0.56)	−0.25 (0.77)	0.34 (*t*-test)
Δ Parathormon, pg/mL, mean (SDS)	6.65 (14.15)	−6.47 (22.22)	0.12 (*t*-test)
Δ Vitamin D, ng/mL, mean (SDS)	−3.06 (11.22)	2.03 (10.41)	0.3 (*t*-test)
Δ Neutrophils, ×10^3^/µL, mean (SDS)	0.27 (1.44)	−0.36 (2.24)	0.45 (*t*-test)
Δ White Blood Cells, ×10^3^/µL, mean (SDS)	−0.16 (1.45)	−0.91 (2.13)	0.38 (*t*-test)
Δ Platelets, ×10^3^/µL, mean (SDS)	−20.09 (41.49)	−32.6 (34.67)	0.47 (*t*-test)
Δ Microalbuminuria mg/L, mean (SDS)	152.6 (151.0)	38.9 (39.56)	0.03 (*t*-test)
Δ Glycemia, mg/dL, mean (SDS)	0.67 (7.86)	3.7 (15.26)	0.56 (*t*-test)
Δ GOT, U/L, mean (SDS)	−1.45 (8.08)	−7.2 (8.47)	0.13 (*t*-test)
Δ GPT, U/L, median (IQR)	1.0 (11.0)	−4.5 (6.0)	0.13 (Mann-Whitney-test)
Δ Na^+^, mEq/L, mean (SDS)	1.09 (1.7)	−0.4 (2.67)	0.14 (*t*-test)
Δ K^+^, mEq/L, mean (SDS)	−0.18 (0.48)	−0.17 (0.35)	0.95 (*t*-test)
Δ Cl^−^, mEq/L, mean (SDS)	0.82 (2.18)	−0.6 (2.59)	0.19 (*t*-test)
Δ Ferritin, µg/L, mean (SDS)	2.55 (15.27)	1.1 (8.99)	0.80 (*t*-test)
Δ Total cholesterol, mg/dL, median (IQR)	−4.5 (6.0)	1.0 (11.0)	0.46 (Mann-Whitney-test)
Δ Triglycerides, mg/dL, mean (SDS)	30.64 (55.64)	−0.6 (52.89)	0.20 (*t*-test)
Δ KIDMED score, mean (SDS)	−1.4 (1.5)	−0.1 (0.90)	0.03 (*t*-test)
Δ Physical activity, hours/week, mean (SDS)	−4.05 (6.53)	−4.35 (4.19)	0.90 (*t*-test)
Δ Screen time, hours/week, mean (SDS)	17.82 (8.49)	13.85 (8.53)	0.30 (*t*-test)
Δ Sweets/Candies, n/week, mean (SDS)	3.5 (3.4)	0.7 (0.9)	0.02 (*t*-test)

Abbreviations: BMI, body mass index; CKD, chronic kidney disease; DBP, diastolic blood pressure; Δ, delta; GOT, glutamate-oxalacetate transaminase; GPT, glutamate-pyruvate transaminase; eGFR, estimated glomerular filtration rate; IQR, interquartile range; KIDMED, Mediterranean Diet Quality Index for Children and Adolescents questionnaire; SBP systolic blood pressure; SDS, standard deviation score; UPr/Cr, urinary protein/creatinine ratio.

## Data Availability

The datasets generated during and/or analysed during the current study are available from the corresponding author on request.

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
