# Peer review of "Effects of Lockdown for COVID-19 Pandemic on Chronic Kidney Disease Progression in Children with Congenital Anomalies of the Kidney and Urinary Tract: A Retrospective Pilot Study"

_children, 2023, doi:10.3390/children10010123_

Round 1

Reviewer 1 Report

Dear Authors,

It is an important topic. I have some comments.

You should delete the dot in the title. You should indicate the study’s design with a commonly used term in the title. You may find better title for your study.

You should explain your patient pool size. How many children and their age group. Do you use any sample selection method?

How to prepare your survey? Dou you use any references for your survey? Which language did you use? Did you do a pilot study for questionnaire? How to prevent recall bias?

Did you follow any guideline for reporting? You may find in https://www.equator-network.org/

Which version of the KIDMED questionnaire did you use, Italian or English?

You should use reference for validity of KIDMED.

How to evaluate physical activity level? Did you use any valid physical activity questionnaire?

How to evaluate sweets or candies they ate in a week? Did you use any portion sizes?

You should add dates about when did you collected your survey.

You should check your statistical analysis. Because you should use paired sample t-test or nonparametric two-related samples test.

I recommend getting statistics support.

You should explain how to give this p-values in Table 1. What is the meaning of this p-values?

Stage 1 CKD, No. (%) 7 (33.3)     5 (23.8)               0.73

Stage 2 CKD, No. (%) 12 (57.1)   13 (61.9)            0.99

Stage 3 CKD, No. (%) 1 (4.8)       2 (9.5)                 0.99

Stage 4 CKD, No. (%) 1 (4.8)       1 (4.8)                 0.99

Stage 5 CKD, No. (%) 0 (0)           0 (0)                    0.99

You should add explanations below the tables about which test you are using.

You should use “sex” instead of “gender”.

“For example” is not formal.

It would not be correct to state that the changes were only in CKD patients without a control group.

You should explain what is the direct or indirect effects of the lockdown among CKD patients.

You should discuss limitations of the study, taking into account sources of potential bias or imprecision. Discuss both direction and magnitude of any potential bias.

Discuss the generalisability of the study results.

Best wishes

Author Response

Dear Authors, it is an important topic. I have some comments.

  1. You should delete the dot in the title. You should indicate the study’s design with a commonly used term in the title. You may find better title for your study. 

Answer: we indicated in the title that this is a retrospective pilot study. Please see line 4 of the new version of the manuscript.

  1. You should explain your patient pool size. How many children and their age group. Do you use any sample selection method? 

Answer: 5 patients had an age between 3 and 5 years, 5 patients between 6 and 10 years, 6 between 10 and 15 years, and 5 between 16 and 18 years (please see lines 210-212 of the new version of the manuscript). Our sample size was inficiated by the fact that patients to be enrolled needed of a follow-up visit within 3 months before the start of the first Italian lockdown (pre-lockdown observation between December 2019 and February 2020) and observed again for a regular follow-up within 6 months after the end of that lockdown (post-lockdown observation between May 2020 and November 2020). 

After the evaluation of the historical cohort, as suggested by the Academic Editor, we were also able to calculate the post-hoc power of this study. We added this information in th new version of the manuscript, please see lines 179-183 of the new version of the manuscript.

  1. How to prepare your survey? Dou you use any references for your survey? Which language did you use? Did you do a pilot study for questionnaire? How to prevent recall bias? 

Answer: thank you for your question. We did not do a pilot study for questionnaires but we used referenced questionnaires. In the new version of the manuscript we added detailed information about used references, the used language and the method to prevent the recall bias. We largely edited the methods section to improve it accordingly with your suggestions. Please see lines 92-136 of the new version of the manuscript.

  1. Did you follow any guideline for reporting? You may find in https://www.equator-network.org/ 

Answer: we used the STROBE reporting guidelines.

  1. Which version of the KIDMED questionnaire did you use, Italian or English?

Answer: all the interviews were made in Italian language. We added this information in the new version of the manuscript (please see line 99).

  1. You should use reference for validity of KIDMED.

Answer: we added the references. Please see lines 131-136 of the new version of the manuscript.

  1. How to evaluate physical activity level? Did you use any valid physical activity questionnaire?

Answer: following this question and your question n°3 we added more information in the text and we largely edited the methods section. Please see lines 92-136 of the new version of the manuscript.

  1. How to evaluate sweets or candies they ate in a week? Did you use any portion sizes?

Answer: following this question and your question n°3 we added more information in the text and we largely edited the methods section. Please see lines 92-136 of the new version of the manuscript.

  1. You should add dates about when did you collected your survey.

Answer: we added this information in the new version of the manuscript. Please see line 84 of the new version of the manuscript.

  1. You should check your statistical analysis. Because you should use paired sample t-test or nonparametric two-related samples test. I recommend getting statistics support.

Answer: according with your comments and to the comments of Reviewer 2 we carefully revised the statistical analysis and we modified it according to your comments (Please see lines 185-283 of the new version of the manuscript).

  1. You should explain how to give this p-values in Table 1. What is the meaning of this p-values?

Stage 1 CKD, No. (%) 7 (33.3)     5 (23.8)               0.73

Stage 2 CKD, No. (%) 12 (57.1)   13 (61.9)            0.99

Stage 3 CKD, No. (%) 1 (4.8)       2 (9.5)                 0.99

Stage 4 CKD, No. (%) 1 (4.8)       1 (4.8)                 0.99

Stage 5 CKD, No. (%) 0 (0)           0 (0)                    0.99

Answer: we added the test used to calculate the p in the new version of the manuscript (please see the new version of the Table 1). Because to evaluate the CKD progression we also considered the drop in estimated glomerular filtration rate by a 25% or more for patients belonging to CKD stage 1 and 2 (please see lines 166-177), we added this information in the Table 1 to give an idea to the Readers of the distribution of CKD stages both at pre- and post-lockdown observation.

  1. You should add explanations below the tables about which test you are using.

Answer: we added the test used in each row of each table.

  1. You should use “sex” instead of “gender”.

Answer: we modified the text accordingly.

  1. “For example” is not formal.

Answer: we deleted “For example”.

  1. It would not be correct to state that the changes were only in CKD patients without a control group. 

Answer: we agree with you, in the new version of the discussion we limited the discussion only to the findings related to the differences between patients with and without CKD progression. We specified the setting of our study also in the first sentence of the discussion in order to not provide misleading information to the Readers. Please see lines 284-343 of the new version of the manuscript.

  1. You should explain what is the direct or indirect effects of the lockdown among CKD patients.

Answer: we modified the text accordingly. Please see lines 326-328 of the new version of the manuscript.

  1. You should discuss limitations of the study, taking into account sources of potential bias or imprecision. Discuss both direction and magnitude of any potential bias.

Answer: we discussed it al lines 314-325 of the new version of the manuscript.

  1. Discuss the generalizability of the study results.

Answer: we discussed it at lines 323-325 of the new version of the manuscript.

Reviewer 2 Report

The paper addresses an interesting issue concerning the impact of lockdown for COVID-19 pandemic on  progression of renal failure in children with CKD. The paper is interesting, however, some issues/data/analyses require to be corrected or supplemented to provide useful and clear information for readers.

Specific comments:

·         (Table 1) Since the data presented in the Table 1 is pre- and post-lockdown for the same subjects, the paired t test (parametric data) or Wilcoxon matched paired test (non-parametric data) should be applied.

·          (Tables) Non-parametric data should be presented as median and percentiles, and not as mean and SD.

·          (Tables) Please indicate for each of parameter which test (t test, Mann-Whitney test, etc.) was used.

·         Data presented in Table 3 suggests that in both, the “worsening” and “non-worsening” populations, there was a shift of patients towards lower categories in KIDMED score after lockdown, that is not consistent with the result of delta KIDMED score presented in Table 2. Could you explain the reason?

·         Authors state that “All the patients presented increased BMI (BMI), reduced physical activity and increased screen time at post- compared with pre-lockdown observation.” Please specify if the change for these parameters was really for each of examined patient or if you meant the result of statistical analysis, that there was a significant difference for mean BMI, physical activity and screen time between pre- and post-lockdown observations.

·         The conclusion of the paper presented in abstract and discussion section is not supported by the obtained results and should be redrafted. Authors did not present in the paper the data confirming that the less adherence to the MD and a major consumption of sweets or candies during lockdown were directly associated to CKD progression or that they were “significant determinants of the CKD progression”.

·         It would be valuable to present the applied questionnaire and the results for the individual questions in the supplemental material.

·         It would be valuable to add BMI results (kg/m2) in the tables, and eGFR values in the Table 2.

·         One of the limitation of the study can be the way of collecting data using phone questionnaire. Please consider this issue in the limitation of study section.

Author Response

The paper addresses an interesting issue concerning the impact of lockdown for COVID-19 pandemic on  progression of renal failure in children with CKD. The paper is interesting, however, some issues/data/analyses require to be corrected or supplemented to provide useful and clear information for readers.

Specific comments:

  1. (Table 1) Since the data presented in the Table 1 is pre- and post-lockdown for the same subjects, the paired t test (parametric data) or Wilcoxon matched paired test (non-parametric data) should be applied. 

Answer: thank you for this suggestion. Following your comment, the analysis and data presentation significantly improved. We applied the paired t test for parametric data and Wilcoxon matched paired test for non-parametric data in the Table 1 and we modified the text accordingly. Please see lines 196-199, 200-202, 259-262, and the Table 1 of the new version of the manuscript.

  1. (Tables) Non-parametric data should be presented as median and percentiles, and not as mean and SD. 

Answer: we modified the text accordingly, please see lines 200-202, 224-226, and 245-247 of the new version of the Tables 1 and 2.

  1. (Tables) Please indicate for each of parameter which test (t test, Mann-Whitney test, etc.) was used.

Answer: we specified this information in the new version of the Tables.

  1. Data presented in Table 3 suggests that in both, the “worsening” and “non-worsening” populations, there was a shift of patients towards lower categories in KIDMED score after lockdown, that is not consistent with the result of delta KIDMED score presented in Table 2. Could you explain the reason?

Answer: thank you for this comment. As shown in the table 1, all the patients presented a worsening of the KIDMED score comparing the pre- and post-lockdown values. However, as shown in the Table 2, the patients showing the CKD worsening presented a higher delta in the reduction of KIDMED score compared with patients without CKD worsening. In the Table 3 are shown the KIDMED categories divided on the basis of intervals of values. In each interval there is a range of values allowing that a people with a KIDMED score of 4 is in the same category of a patient with a KIDMED score of 7. Therefore, also if patients with CKD worsening have presented a higher KIDMED score delta and despite a patient with absolute lower KIDMED score undoubtfully presents a less adherence to Mediterranean Diet, the patients with CKD worsening resulted in the same KIDMED category of patients without CKD progression. We are agree with you that this table could be misleading for the Readers and, because it does add useful information to the paper, we decided to delete it.

  1. Authors state that “All the patients presented increased BMI (BMI), reduced physical activity and increased screen time at post- compared with pre-lockdown observation.” Please specify if the change for these parameters was really for each of examined patient or if you meant the result of statistical analysis, that there was a significant difference for mean BMI, physical activity and screen time between pre- and post-lockdown observations.

Answer: We clarified this issue in the new version of the manuscript. Please see lines 30-34 and 205-268 of the new version of the manuscript.

  1. The conclusion of the paper presented in abstract and discussion section is not supported by the obtained results and should be redrafted. Authors did not present in the paper the data confirming that the less adherence to the MD and a major consumption of sweets or candies during lockdown were directly associated to CKD progression or that they were “significant determinants of the CKD progression”. 

Answer: we modified the conclusions of both abstract and discussion according with your comment. Please see lines 39-42 and 326-343 of the new version of the manuscript).

  1. It would be valuable to present the applied questionnaire and the results for the individual questions in the supplemental material.

Answer: following your comment and the comment n°3, 5, 6, 7, 8 of the Reviewer 1, we added detailed information about the adopted questionnaires in the new version of the methods section. Please see lines 92-136 of the new version of the manuscript.

  1. It would be valuable to add BMI results (kg/m2) in the tables, and eGFR values in the Table 2.

Answer: we added the required values in the Table 2.

  1. One of the limitation of the study can be the way of collecting data using phone questionnaire. Please consider this issue in the limitation of study section.

Answer: following your comment and the comments of Reviewer 1 we largely edited the limitations section within the discussion (Please see lines 314-325 of the new version of the manuscript).

Round 2

Reviewer 1 Report

Dear Author,

Thank you for your all corrections.

It is publishable paper for me.

Best wishes

Author Response

Thank you